# Temperature-Dependent Alternative Splicing of Precursor mRNAs and Its Biological Significance: A Review Focused on Post-Transcriptional Regulation of a Cold Shock Protein Gene in Hibernating Mammals

**DOI:** 10.3390/ijms21207599

**Published:** 2020-10-14

**Authors:** Takahiko Shiina, Yasutake Shimizu

**Affiliations:** Laboratory of Veterinary Physiology, Department of Joint Veterinary Medicine, Faculty of Applied Biological Sciences, Gifu University, 1-1 Yanagido, Gifu 501-1193, Japan; yshimizu@gifu-u.ac.jp

**Keywords:** alternative splicing, cold shock protein, dominant negative, hibernation, temperature

## Abstract

Multiple mRNA isoforms are often generated during processing such as alternative splicing of precursor mRNAs (pre-mRNA), resulting in a diversity of generated proteins. Alternative splicing is an essential mechanism for the functional complexity of eukaryotes. Temperature, which is involved in all life activities at various levels, is one of regulatory factors for controlling patterns of alternative splicing. Temperature-dependent alternative splicing is associated with various phenotypes such as flowering and circadian clock in plants and sex determination in poikilothermic animals. In some specific situations, temperature-dependent alternative splicing can be evoked even in homothermal animals. For example, the splicing pattern of mRNA for a cold shock protein, cold-inducible RNA-binding protein (CIRP or CIRBP), is changed in response to a marked drop in body temperature during hibernation of hamsters. In this review, we describe the current knowledge about mechanisms and functions of temperature-dependent alternative splicing in plants and animals. Then we discuss the physiological significance of hypothermia-induced alternative splicing of a cold shock protein gene in hibernating and non-hibernating animals.

## 1. Introduction

Alternative splicing of precursor mRNAs (pre-mRNAs) is a process in which exons, parts of exons, and/or parts of introns are combinatorially included into mature RNA [1,2,3,4,5,6]. As a result of the process, multiple mature mRNAs are generated from a single gene [7,8,9]. Proteins translated from alternatively spliced mRNA isoforms are different from the authentic protein in structure, function, and thus various properties including intracellular localization, enzymatic activities, binding behaviors and channel functions [5]. In some cases, alternative splicing creates not only a functionally active variant but also an inactive one, and the inactive variant depresses the functions of the active variant (so-called dominant negative effect) [5]. It is also possible that some alternatively spliced transcripts are non-coding but modulate other RNAs by competing with them for their regulators [1]. Hence, alternative splicing is an essential mechanism for the functional complexity of eukaryotes.

Splicing patterns can be changed in response to environmental factors including temperature. Temperature is a fundamental physical quantity that is involved in all life activities at various levels [10]. In plants, environmental temperature affects survival, growth, and fitness [11]. Since cellular temperature in plants would directly reflect the environmental temperature, it can be expected that temperature-dependent changes in alternative splicing patterns play a key role in these adaptive responses. In contrast, the temperature of mammalian organs such as the heart and brain is maintained within a narrow range by a homeostatic mechanism [12]. Thus, it seems reasonable to assume that temperature-dependent regulation of alternative splicing is largely indispensable for mammalian cells. However, in some specific situations, temperature-dependent alternative splicing can be evoked even in homothermal animals. For example, we found that the splicing pattern of mRNA for a cold shock protein, cold-inducible RNA-binding protein (CIRP or CIRBP), is changed in response to a marked drop in body temperature during hibernation of hamsters [13,14]. The temperature-dependent splicing regulation of CIRP transcripts was initially considered to be specific for hibernating animals but was later shown to occur commonly in rats and mice, which are non-hibernators [15]. Here, we review the current knowledge about mechanisms and functions of temperature-dependent alternative splicing in plants and animals. Then we discuss the physiological significance of hypothermia-induced alternative splicing of cold shock protein gene in hibernating and non-hibernating animals.

## 2. Regulation of Gene Expression by Temperature-Dependent Alternative Splicing

Multiple mRNA isoforms are often generated by alternative splicing of pre-mRNA, resulting in the diversity of generated proteins [7,8,9]. Various mechanisms for the regulation of alternative splicing are known [1,3,4,6]. Several studies have demonstrated that changes in temperature affect pre-mRNA splicing [16,17,18]. Temperature-dependent alternative splicing is involved in various functions in both plants and animals (Table 1). In many cases, regulatory systems in plants such as flowering and circadian clock has been reported. There are also reports about temperature-dependent alternative splicing in animals including mammals.

### 2.1. Flowering

In response to changes in ambient temperature, plants show multiple phenotypes such as stem elongation, leaf hyponasty, and flowering [19,20,21,22]. These thermal responses are regulated by various genes [22]. Among the various phenotypes, thermal control of flowering has been extensively studied because it is directly associated with reproductive success and crop productivity [21,23]. Researches on the regulation of flowering have indicated that endogenous and exogenous factors may be integrated to create a developmental switch in plants [24,25]. In recent decades, key regulatory molecules of flowering and their targets have been identified in Arabidopsis [25].

Flowering time in plants is regulated in response to change in ambient temperature [16,26]. The temperature-dependent regulation is associated with alternative splicing of flowering-related genes [16]. Several regulators for flowering time have been shown to be controlled by temperature-dependent alternative splicing. One of the regulators that have been characterized in detail is flowering locus M (*FLM*), which is a MADS-domain transcription factor [16,23,27,28]. *FLM* is subjected to temperature-dependent alternative splicing in Arabidopsis [23,27,28,29,30,31,32,33,34,35,36]. Two splice variants of *FLM*, *FLM-β* and *FLM-δ*, differ in the incorporation of either the second or third cassette exon [11,27,37]. These splice variants are oppositely regulated by temperature and are translated into proteins with opposing functions, acting as a repressor (*FLM-β*) and as a promoter (*FLM-δ*) of flowering [11,23,27,37]. Intriguingly, the ratio of *FLM-β* to *FLM-δ* alters in response to ambient temperature [25,29]. At a low ambient temperature, *FLM-β* is expressed predominantly and *FLM-β* forms a protein complex with the MADS-domain transcription factor short vegetative phase (SVP), which also functions as a floral repressor [21,23] and can bind the target DNA to actively repress flowering [21,27,35]. On the other hand, at a high ambient temperature, *FLM-δ* is spliced abundantly and *FLM-δ* interacts with SVP and works as a dominant-negative isoform of *FLM* [23,25,27]. The SVP–FLM-δ complex cannot bind DNA [21,23,27], resulting in the release of floral promoters such as flowering locus T (FT) and suppressor of overexpression of constans 1 (SOC1) from repression and thereby indirectly inducing flowering at elevated temperatures [23,27]. Collectively, these observations indicate that alternative splicing of *FLM* is a critical molecular action for the *FLM*-mediated control of flowering in response to thermal change [21]. 

*MADS affecting flowering 2 (MAF2)*, which is related to flowering time like *FLM* in Arabidopsis, also undergoes temperature-dependent alternative splicing to produce two variants, *MAF2var1* and *MAF2var2* [11,25,38,39,40,41]. *MAF2var1* is expressed predominantly at a low ambient temperature and its encoded protein forms a repressive complex with SVP to inhibit flowering [11,38,40,41,42]. At high ambient temperature, expression of the intron-retaining splice variant *MAF2var2* is induced [38,40,42]. The truncated protein produced from *MAF2var2* cannot interact with SVP, resulting in promotion of flowering [11].

### 2.2. Circadian Clock

The circadian clock produces oscillations of an approximately 24-h period, which allows organisms to synchronize biological phenomena with the day-night cycle [16,43]. In plants, the circadian clock harmonizes physiological processes including rhythmic leaf movements, stomatal aperture and flowering [16,25,44,45,46,47]. The circadian clock is synchronized by light and temperature inputs [48]. In Arabidopsis, the circadian clock is composed of several interlocking regulatory feedback loops [25,46]. Circadian clock associated 1 (CCA1) and late elongated hypocotyl (LHY), MYB transcription factors that are expressed in the morning, repress the activity of the evening complex, which is composed of early flowering (ELF)3, ELF4, and lux arrhythmo (LUX) [16,25,49,50]. The evening complex represses *pseudo response regulator (PRR)* genes, including *PRR7*, *PRR9* and *timing of CAB expression 1 (TOC1; PRR1*) [16,44,46,51]. The PRR factors repress CCA1 and LHY [25,51,52,53]. In addition, CCA1 and LHY repress the expression of *TOC1* through direct binding to its promoter [25,54,55]. 

There have been many reports on post-transcriptional regulatory mechanisms of circadian clock genes [16,25,56]. Circadian clock genes are subjected to temperature-dependent alternative splicing [16,25]. Alternative splicing of *CCA1* pre-mRNA produces two splice variants, *CCA1α* and *CCA1β* [57,58,59,60,61]. The roles of these variants in regulation of the circadian clock have been reported [61]. At a high temperature, *CCA1β* is increased. CCA1β has a dimerization domain but lacks the N-terminal DNA-binding MYB motif [61,62]. Thus, CCA1β represses CCA1α/LHY heterodimerization by competing with CCA1α and LHY to form non-functional protein complexes such as CCA1α/CCA1β and CCA1β/LHY heterodimers as a dominant regulator [61,63]. In contrast, at a low temperature, CCA1β is decreased, promoting functional CCA1α/LHY heterodimerization [58,61,64]. 

In addition to *CCA1*, the circadian clock genes *LHY*, *TOC1*, *PRR3*, *PRP5*, *PRP7,* and *PRR9* are also subjected to temperature-dependent alternative splicing [59,64]. James et al. identified many novel alternative splicing events including the addition of exon 5a in *LHY*, exon 4 skip in *PRR7*, retention of intron 4 in *TOC1*, and three and two alternative splicing events in *PRR3* and *PRR5*, respectively [59]. Several of these changes are dependent on temperature transitions and affect clock gene expression by producing nonfunctional transcripts and/or inducing nonsense-mediated decay (NMD) [59]. For example, temperature-dependent alternative splicing contributes to down-regulation of *LHY* transcript at low temperature [59]. Calixto et al. demonstrated changes in alternative splicing of *LHY* and *photoperiod-H1* (*PPD-H1*) (*PRR7* orthologue) in response to a low temperature in barley [65]. In addition, heat stress significantly upregulates a splice variant of *PRR7* that retains an intron, whereas *PRR9* shows an increase in both full length mRNA and a splice variant with an intron retention after both cold and heat stresses [16,64]. 

In addition to investigations of circadian clock genes in plants, it has also been shown that temperature-dependent alternative splicing is essential for the regulation of circadian clock in Neurospora, Drosophila, tilapia and mice [66,67,68,69,70,71,72,73,74,75,76]. These findings suggest that alternative splicing is a general mechanism of clock regulation. In Neurospora, the circadian clock gene *frequency (FRQ)* is subjected to temperature-dependent alternative splicing [67,68,69]. Expression levels and ratios of the long (l) and short (s) isoforms of *FRQ* are important for temperature compensation of circadian rhythms. Diernfellner et al. demonstrated that the ratio of *l-FRQ* versus *s-FRQ* is regulated by temperature-dependent alternative splicing of intron 6 of *FRQ* and then expression of *l-FRQ* increases with elevating temperature, while *s-FRQ* is expressed at low levels [69]. 

In Drosophila, *period (PER)* and *timeless (TIM)* protein and mRNA products undergo daily fluctuations in relation to the circadian clock [66,71,73,74,77]. Majercak et al. showed that temperature-dependent alternative splicing in the *PER* gene plays a key role in adaptation of a circadian clock to seasonally cold days (low temperatures and short day lengths) in Drosophila [73]. The enhanced splicing at low temperatures advances the steady-state phases of *PER* mRNA and protein cycles, which contributes to the preferential daytime activity of flies on cold days [73]. Martin Anduaga et al. showed that temperature dramatically changes the splicing pattern of *TIM* in Drosophila [74]. At 18 °C, *TIM* levels are low because of the induction of two cold-specific isoforms, while another isoform is upregulated at 29 °C. The switching of isoform controls the levels and activity of TIM, suggesting that alternative splicing of *TIM* might act as a thermometer for the circadian clock [74]. In addition to Drosophila, in Nile tilapia, alternative splicing in the *PER* gene is also affected by cold stress [72].

Alternative splicing of circadian clock genes has been demonstrated in mice. For example, alternative splicing in *U2 snRNP auxiliary factor (U2AF) 26* gene is driven by rhythmic body temperature changes, and then skipping of two variable exons (6 and 7) of *U2AF26* generates a novel protein variant of U2AF26 that directly controls circadian clock resetting in vivo [70,75,76]. 

Taken together, these findings indicate that many clock-related genes are subjected to temperature-dependent alternative splicing. It has been speculated that alternative splicing might contribute to the control of circadian clock functions in a temperature-dependent manner.

### 2.3. Sex Determination

Sex determination is a pivotal developmental process [78]. There are two mechanisms in sex determination. One is based on sexual genetic elements, which is called genotypic sex determination (GSD). The other is based on environmental factors, known as environmental sex determination (ESD) [78,79,80]. Temperature-dependent sex determination (TSD) is the most common ESD mechanism in vertebrates, in which individuals become male or female depending on the incubation temperature that the developing embryo experiences [81]. TSD is found in many vertebrates including all crocodilians and tuatara, most turtles, some lizards and some fish [78,81]. Some genes including *SRY-box9* (*Sox9*), *Wilms tumor 1*(*WT1*) and *doublesex and mab3-related transcription factor 1* (*Dmrt1*) are involved in TSD [78,82,83,84,85,86]. Yamaguchi and Iwasa pointed out that temperature-dependent alternative splicing is a possible biological mechanism for temperature sensitivity [87]. In accordance, *Sox9* in crocodiles, *WT1* in turtles, and *Dmrt1* in turtles and crocodiles are subjected to alternative splicing and play a role in sex determination [78,82,83,84,86]. For example, Rhen et al. found that a change in temperature rapidly influences the expression and splicing of *WT1* mRNA, which influences sex determination in the snaping turtle [84]. 

### 2.4. Others

Temperature can modulate the splicing of splicing regulators. For example, *U2AF65A* [88], *cyclin-dependent kinase G1* (*CDKG1*) [33,88], *SR1* [89], *polypyrimdine tract-binding protein* (*PTB*) *1/2*, and *suppressor of abi3-5* (*SUA*) [90], which affect the splicing pattern, are subjected to temperature-dependent alternative splicing in Arabidopsis. In animals, the splicing of pre-mRNA of *U2AF26* in mice [70,75,76] and *serine-arginine-rich (SR) splicing factors* in Tilapia [72] are also regulated by change in temperature. It is interesting that splicing itself contributes to the control of splicing.

Genes related to plant growth and development are also subjected to temperature-dependent alternative splicing. In Arabidopsis, it has been demonstrated that temperature-dependent alternative splicing of *root initiation defective1* (*RID1*) [91] and *apetala3* (*AP3*)*-1* [92] affects plant development and floral phenotype.

Most of the temperature-dependent alternative splicing studies in plants have focused on Arabidopsis. However, there are several reports about plants other than Arabidopsis. It was shown that temperature-dependent alternative splicing of *OsbZIP58* affects grain filling of rice [93]. In sugar beet, *Bvnpcg2* and *Bvnpcg3* are subjected to temperature-dependent alternative splicing, resulting in regulation of the sensitivity to seasonal temperature changes [94]. In Medicago, temperature-dependent alternative splicing of *MtJMJC5,* a jumonj C domain-containing demethylase homologue, is involved in the circadian clock [95]. 

Although mammals have thermosensory molecules such as family of transient receptor potentials (TRPs) [96,97,98,99], plants do not possess orthologous molecules that act as specialized thermosensors [16]. Temperature-dependent alternative splicing might therefore function as a “thermometer” in plants to measure changes in ambient temperature [16,23,25,27]. In Arabidopsis, tea plants and Jujuncao, RT-PCR and RNA sequence analyses revealed that alternative splicing of a variety of genes is changed dramatically during cold acclimation [100,101,102,103]. The findings suggest that alternative splicing events play an important regulatory role in response to cold stress [100,101,102,103,104]. 

GrpE proteins function as nucleotide exchange factors for DnaK-type Hsp70s, which is a molecular chaperone [105,106,107]. Schroda et al. identified *CGE1* as a chloroplast homolog of *GrpE* in Chlamydomonas and showed that temperature-dependent alternative splicing generates two isoforms of *CGE1*, *CGE1a* and *CGE1b* [107,108]. *CGE1a* is expressed predominantly at lower temperatures, but expression of *CGE1b* increases as well as *CGE1a* in response to elevated temperatures, which might affect chaperone function [107].

In addition to plants and animals, temperature-dependent alternative splicing has been reported in yeast [109,110,111]. In a search for yeast transcripts, splicing of *APE2*, which encodes an amino peptidase, is affected by temperature change (heat shock) [109,111].

Several studies on temperature-dependent alternative splicing of pre-mRNAs have been performed by using mammalian cultured cells. Gemignani et al. used HeLa and K562 cell lines that stably express thalassemic β-globin genes with mutations inducing aberrant splicing patterns of pre-mRNAs [112]. When the cells were cultured at temperatures below 30 °C, aberrant splicing was inhibited and correct splicing was restored. The findings suggest the possibility of lowering the temperature of bone marrow as a treatment for β-thalassemia. Weil et al. examined alternative splicing of a collagen gene in fibroblasts of a patient with Ehlers-Danlos syndrome [113]. Analysis of RT-PCR products showed that fibroblasts grown at 37 °C produced normally spliced and misspliced mRNAs, while only correctly spliced products were detectable when the cells were cultured at 31°C. Tzani et al. performed RNA sequencing in Chinese hamster ovary (CHO) cells and demonstrated that a temperature shift induces alternative splicing of *Dnml1* and *Mff* genes [114]. 

## 3. Molecular Mechanism of Temperature-Dependent Alternative Splicing

Although many genes regulated by temperature-dependent alternative splicing have been identified, molecular mechanisms of temperature-dependent alternative splicing are poorly understood. Temperature can influence the speed of RNA polymerase II, which is known to affect splicing [16,115]. In addition, temperature can affect RNA structure [16,109,110,116]. The secondary structure of intron-exon boundaries affects the spliceosome complex that performs the removal of introns and the joining of exons [109,110]. In addition, the secondary structure of splicing enhancer and silencer motifs also is concerned with splice site selection [116]. These factors might contribute to the regulation of temperature-dependent alternative splicing. 

Temperature-dependent splicing might also be involved in more complex regulatory steps than direct temperature-driven changes in RNA secondary structure [110]. For example, Preußner et al. showed that body temperature cycles drive rhythmic phosphorylation of SR proteins to regulate alternative splicing in mice [75]. A temperature change of 1 °C is sufficient to induce a concerted splicing switch in many genes that are functionally related to circadian rhythm. In addition, Haltenhof et al. showed that a lower body temperature activates CDC-like kinases (CLKs), resulting in strongly increased phosphorylation of SR proteins [117]. This globally controls temperature-dependent alternative splicing and gene expression with wide implications in circadian, tissue-specific, and disease-associated settings [117].

It is interesting that temperature can modulate the splicing of splicing regulators such as SR proteins [16,89,118,119]. Splicing regulators are themselves subjected to temperature-dependent alternative splicing, affecting the splicing pattern of a large number of genes. James et al. investigated the role of RNA-binding splicing factors (SFs) in temperature-dependent alternative splicing of *LHY* in Arabidopsis. They showed that the splicing and expression of several SFs respond sufficiently, rapidly, and sensitively to temperature changes to contribute to the splicing of *LHY* pre-mRNA [90]. 

Melzer reported that the presence/absence of SFs may contribute to alternative splicing [120]. He also pointed out that temperature-dependent RNA structures (symbolized by the presence/absence of the stem-loop structure) are involved, although it is not yet entirely clear how plants sense temperature differences.

Foley et al. demonstrated that the alternative splicing regulator P-element somatic inhibitor (PSI) regulates the temperature-dependent alternative splicing of *TIM* in Drosophila, regulating the period of circadian rhythms and circadian behavior phase during temperature cycling [77]. 

It has been reported that trimethylated histone H3 at lysine 36 (H3K36me3) is enriched in genes undergoing alternative splicing in mammals [21,121,122]. In addition, it has been shown that the H3K36me3-enriched genomic sequence regions, which include flowering-related genes, are broader at warm temperatures in Arabidopsis exposed to different ambient temperatures [123]. These findings indicate that H3K36me3 might mediate the temperature-dependent alternative splicing of flowering-related genes [21]. 

## 4. Temperature-Dependent Regulation of Alternative Splicing in Gene Expression of CIRP

### 4.1. Discovery of Temperature-Dependent Alternative Splicing of a CIRP Transcript during Hibernation in Hamsters

Some mammalian species including squirrels and hamsters undergo hibernation [124,125]. Although body temperature drops to less than 10 ºC, the heart of hibernating animals can maintain constant beating [124]. This is in contrast to non-hibernators, in which the heart cannot maintain beating in a deep hypothermic condition [126]. Therefore, the hearts of hibernating animals would possess a protective mechanism against the harmful effects of a low temperature [127]. The innate ability of the heart that is specific for hibernators definitely contributes to the cold-resistant nature. However, maintenance of normal cardiac function under a condition of deep hypothermia would not exclusively depend on the innate ability. In fact, the heart of hamsters is damaged when deep hypothermia is forcibly induced by pentobarbital anesthesia and cooling [128]. Accordingly, it is most probable that some mechanisms actively operate during hibernation to protect the heart and also other organs.

Originally, we were interested in the protective mechanisms operating in hibernating animals. As a molecular candidate involved in the mechanism, we focused on cold shock proteins including as CIRP and RNA-binding motif 3 (RBM3) [129]. This is because the characteristics of the proteins fit well to our working hypothesis as follows. Firstly, expression levels of CIRP and RBM3 are increased by cold stress [129,130,131]. Secondly, these proteins regulate gene expression at the level of translation, enabling cells to respond rapidly to cold stress [129,132]. Finally, CIRP and RBM3 exert protective effects on various types of cells exposed to a harmful low temperature [98,133]. 

Owing to the prominent action of cold shock proteins, we considered that these proteins might help to protect organs including the heart against a harmful low temperature during hibernation. At the beginning of our experiments, we simply expected that cold shock proteins are expressed, if any, at very low levels, and that their expression is increased before or when entering a deep hypothermic state (torpor) in hamsters. However, RT-PCR analysis led to two unexpected results. One result was that there was substantial expression of *CIRP* mRNA in the heart of a non-hibernating euthermic hamster [14]. The other was that the electrophoretic pattern of PCR products amplified from the hearts of hibernating hamsters was clearly different from that of non-hibernating hamsters. In non-hibernating animals, three different *CIRP* mRNAs were detected, whereas only the shortest variant among them was expressed in hibernating animals (referred to as short-form dominant pattern). The pattern of splicing was commonly observed in various organs [13]. These findings allowed us to infer *CIRP* expression is regulated at the level of pre-mRNA splicing, rather than transcriptional level, when entering torpor. At that stage, we believed that the regulation of alternative splicing is specific for hibernators because it is inconceivable that non-hibernators will be in an extreme hypothermic condition.

### 4.2. Factors Triggering “Hibernation-Specific” Alternative Splicing

We investigated the factors causing the shift in alternative splicing of *CIRP* during hibernation in hamsters. It is generally considered that the cold resistance property is established during preparatory processes before entering hibernation [134]. Therefore, it is rational to anticipate that the shift in alternative splicing of *CIRP* would occur when hamsters were kept in a condition suitable for induction of hibernation. Our experiments, however, showed that the short-form dominant pattern was not induced even after adaptation to a cold environment with short photoperiod [14], which is an experimental condition for induction of hibernation. These experiments also proved that endogenous regulatory pathways activated by signals arising from the body surface (response to cold ambient temperature) and/or those from the eyes (response to reduced light input) play minor roles, if any, in the shift to the short-form dominant pattern.

We then tested whether the splicing pattern can be caused in artificial hypothermia. An adenosine A1-receptor agonist was administrated centrally to induce hibernation-like hypothermia [13,128]. Since activation of adenosine A1-receptors in the brain is an essential process for the decrease in body temperature in natural hibernation [135,136], that method would mimic at least in part the endogenous condition when entering hibernation. Induction of hypothermia by central application of an adenosine A1-receptor agonist reproduced the short-form dominant pattern [13]. This apparently suggests that autonomic and/or endocrine pathways following A1 receptor activation may be responsible for the change in the splicing pattern. However, the agonist was unable to induce the short-form dominant pattern when the drop in body temperature was prevented by warming the body of animals [13]. If adenosine A1-receptors in the central nervous system activate specific regulatory pathways to shift the splicing, the short-form dominant pattern should be observed in the absence of an actual decrease in body temperature. It thus seems likely that autonomic and/or endocrine pathways, which may operate after activation of central adenosine A1-receptors, are not directly involved in the regulation of alternative splicing. Taken together, the findings indicate that reduced temperature is more likely to be the major contributing factor of the shift in alternative splicing during hibernation.

### 4.3. Mild Hypothermia as a Major Cause of the Shift in Alternative Splicing of CIRP Transcripts

To test the hypothesis that reduced temperature is the major cause of the shift in the splicing pattern, we induced hypothermia forcibly in hamsters. Cooling the anesthetized animals successfully induced hibernation-like hypothermia [14,128]. Despite sufficient reduction of body temperature, a “hibernation-specific” pattern of alternative splicing was not reproduced in the forcibly induced hypothermia. At that stage, we noticed a marked difference between adenosine A1- receptor agonist-induced hypothermia and anesthetic-induced hypothermia. The former dropped to 10 ºC within one hour, whereas the latter took about 5 h to reach the same level of hypothermia [13,14] (Figure 1). Based on this difference, we assumed that staying in a mildly hypothermic state would be necessary to shift the alternative splicing of *CIRP* transcripts. If that is the case, the short-form dominant pattern should be reproduced by maintaining hamsters in a state of mild hypothermia for a substantial period of time even in the case of hypothermia induced by anesthetics. Predictably, maintenance of body temperature in a mild hypothermic state (~ 30 °C) for 2 h elicited a shift in the pattern [13]. The short-form dominant pattern generated when staying in a mild hypothermia state can be maintained even after subsequent reduction of body temperature to 10 °C [13]. The findings indicate that it is needed to keep animals in a state of mild hypothermia for a proper period to induce a change in the alternative splicing (Figure 1). This theory can be applied to natural hibernation as well because body temperature decreases gradually in hamsters entering torpor [136,137], being maintained in a state of mild hypothermia for several hours. Based on the results, it is reasonable to conclude that the major factor of the shift in alternative splicing of *CIRP* is mild hypothermia (Figure 1). Therefore, the shift in the alternative splicing pattern of *CIRP* mRNA might not be due to mechanisms specific for hibernators.

### 4.4. Alternative Splicing of CIRP mRNA in a Non-Hibernator

At the final stage of a series of experiments, we examined whether the temperature-dependent alternative splicing of *CIRP* mRNA is actually specific for hibernators or whether similar changes of alternative splicing commonly exist in non-hibernators such as mice. We found that several splicing variants of *CIRP* mRNA are constitutively present in various organs of euthermic mice in a manner similar to that in hamsters [15]. When mild hypothermia is induced in mice, the short-form variant is dominantly expressed, that is, a short-form dominant pattern is generated [15]. In line with this, it has been reported that *CIRP* mRNA is accumulated by changing efficient *CIRP* pre-mRNA splicing in response to a temperature shift in mouse fibroblasts [130,138]. In addition, we found similar temperature-dependent splicing regulation of *CIRP* transcripts in rats (unpublished observation). Therefore, the control system of *CIRP* expression at the level of alternative splicing, which was originally found out in hibernating hamsters, may be commonly present in non-hibernators such as mice.

### 4.5. Hypothesis for the Physiological Significance of Temperature-Dependent Alternative Splicing of CIRP mRNA

As mentioned above, *CIRP* mRNA is expressed in the heart of a non-hibernating euthermic hamster with several different forms probably due to alternative splicing [13,14]. Among them, we focused on the shortest and longest variants (referred as short form and long form, respectively) because of their greater relative abundance. Sequence analysis showed that the short form contains the open reading frame encoding full-length functional CIRP (Figure 2). The long form retains an intron including a stop codon, and then it would be translated to an isoform of CIRP, of which a number of amino acids in the C-terminal region are replaced (Figure 2). It has been reported that the C-terminal arginine-glycine-rich domain is associated with functional activation of CIRP [132,139]. Therefore, it can be postulated that the CIRP isoform, in which C-terminal amino acids are replaced, is not functional. On the other hand, the RNA binding activity of the isoform would remain normal because the RNA-binding domain in the N-terminal region is totally conserved. These structural considerations suggested that the C-terminally replaced isoform plays a dominant-negative role over the full-length CIRP. In accordance, generation of isoforms that have a dominant negative effect on the active isoforms by alternative splicing has been widely demonstrated in several genes encoding enzymes, transporters, channels, or transcription factors [5]. The findings that expression of the splicing variant encoding the putative dominant-negative isoform is greatly decreased and that the variant encoding functional CIRP is predominantly expressed in hibernating animals [13,14] are meaningful for estimating the physiological significance of the temperature-dependent regulation of alternative splicing of *CIRP* transcripts.

Figure 2 illustrates our hypothesis. The most important point is rapid manifestation of the protective effects of CIRP. Under a non-hibernating euthermic condition, expression of the dominant-negative isoform would constitutively compete with the functional CIRP. This may be helpful to mask an unnecessary function during a euthermic active state. Once the decline of body temperature begins, the route for generating mRNA encoding a dominant-negative isoform is shut down, resulting in efficacious production of mRNA for functional CIRP without accompanying transcriptional regulation. It is therefore expected that the shift from the splicing pattern of a euthermic condition to that of a hibernating condition contributes to both the preferential expression of functional CIRP and removal of the masking effect of a dominant-negative isoform. Importantly, the switching of the splicing pattern can be triggered automatically in response to a mildly cold temperature, ensuring its operation at the proper timing. The mechanism of splicing would permit rapid establishment of cold-resistant properties in various organs during gradual reduction of body temperature. 

We believe that the temperature-dependent switching of the splicing pattern plays a significant role in protection of cells against untoward effects of hypothermia, although no definitive evidence has so far been provided. One of the supportive findings is a simultaneous occurrence of the shift in the splicing pattern in most organs. Taking into consideration the fact that the reaction is commonly observed in mice and rats, it is probable that the regulation of *CIRP* expression at the level of splicing is a fundamental event for survival at a low temperature. In hamsters, the occurrence of cardiac arrhythmias and/or cardiac damage is correlated with the pattern of alternative splicing of *CIRP* transcripts (Table 2). Regardless of the method used for inducing hypothermia, a rapid decline in body temperature, which does not cause a shift of splicing, results in abnormal electrocardiograms (ECG) such as J wave and/or signs related to artioventricular block. The origin of the J-wave during hypothermia has been due to injury current, delayed ventricular depolarization and early repolarization, tissue anoxia and acidosis [140]. On the other hand, a slow decline in body temperature promotes the shift of splicing, and normal sinus rhythm can be maintained. More strikingly, induction of deep hypothermia is unsuccessful (i.e., cardiac arrest is induced) when body temperature is decreased in the absence of the shift in splicing of *CIRP* transcripts (Table 2). It is noteworthy that the hearts of mice and rats can keep beating even in a deep hypothermic condition (~ 15 ºC) when the shift in splicing is induced by keeping a state of mild hypothermia before cooling the animals. These findings suggest an indispensable role of the temperature-dependent splicing in protection of cells against a harmful low temperature.

**Table 1 ijms-21-07599-t001:** Temperature-dependent alternative splicing.

Species	Genes	Related Functions	References
Arabidopsis	FLM	Flowering	[23,27,28,29,30,31,32,33,34,35,36]
Arabidopsis	MAF2	Flowering	[11,25,38,39,40,41,42]
Arabidopsis	CCA1	Circadian clock	[57,58,59,60,61]
Arabidopsis	LHY	Circadian clock	[57,59,60,64,90,141]
Arabidopsis	TOC1	Circadian clock	[59,64]
Arabidopsis	PRR3	Circadian clock	[59,64]
Arabidopsis	PRR5	Circadian clock	[59,64]
Arabidopsis	PRR7	Circadian clock	[59,64]
Arabidopsis	PRR9	Circadian clock	[59,64]
Barley	LHY	Circadian clock	[65]
Barley	PPD-H1	Circadian clock	[65]
Neurospora	FRQ	Circadian clock/thermosensing	[67,68,69]
Drosophila	PER	Circadian clock	[71,73]
Drosophila	TIM	Circadian clock	[66,74,77]
Tilapia	PER1/2	Circadian clock	[72]
Mouse	U2AF26	Circadian clock/alternative splicing	[70,75]
Mouse	CIRP	Circadian clock	[130,138]
Crocodile	Sox9	Sex determination	[82]
Turtle	WT1	Sex determination	[84,86]
Turtle	Dmrt1	Sex determination	[78]
Mugger (Crocodile)	Dmrt1	Sex determination	[83]
Arabidopsis	U2AF65A	Alternative splicing	[88]
Arabidopsis	CDKG1	Alternative splicing	[33,88]
Arabidopsis	SR1	Alternative splicing/temperature adaptation	[89]
Arabidopsis	PTB1/2	Alternative splicing	[90]
Arabidopsis	SUA	Alternative splicing	[90]
Tilapia	SR splicing factors	Alternative splicing/temperature adaptation	[72]
Arabidopsis	RID1	Plant development	[91]
Arabidopsis	AP3-1	Floral phenotype	[92]
Rice	OsbZIP58	Grain filling	[93]
Sugar beet	Bvnpcg2/3	Sense seasonal temperature changes	[94]
Medicago	MtJMJC5	Circadian clock	[95]
Arabidopsis		Cold stress response	[100,103]
Tea plant		Cold stress response	[102]
Jujuncao		Cold stress response	[101]
Chlamydomonas	CGE1	Chaperone	[107,108]
Yeast	APE2	Thermosensing	[109,111]
Human (HeLa cell)	β-globin		[112]
Human (fibroblast)	Collagen		[113]
Hamster (CHO cells)	Dnml1/ Mff		[114]
Syrian Hamster	CIRP	Hibernation	[13,14]
Mouse	CIRP	Hypothermia	[15]

## 5. Conclusions and Perspectives

Temperature-dependent alternative splicing of pre-mRNAs is widely conserved in eukaryotes including both plants and animals. The post-transcriptional regulatory system affected by the temperature shift plays important roles in many physiological and developmental processes such as flowering, circadian clock, sex determination, thermosensing, stress response and hibernation. Further studies on temperature-dependent alternative splicing may provide a breakthrough in understanding multiple functions that are associated with adaptation to change in ambient or body temperatures.

We have demonstrated that a shift in the alternative splicing pattern of *CIRP* transcripts is elicited depending on change in body temperature during hibernation [13,14]. Here, we point out that not only the shift of splicing itself but also NMD-dependent degradation of the long form splicing variant, which contains a premature termination codon (PTC), might occur under hibernation or/and hypothermia. In future, it would be useful to examine whether NMD is involved in regulation of gene expression during hibernation. Notably, the change of *CIRP* transcripts is not specific to hibernators but also occurs in non-hibernators. Therefore, the establishment of a method for inducing hibernation-like hypothermia in humans would be feasible.

## Figures and Tables

**Figure 1 ijms-21-07599-f001:**
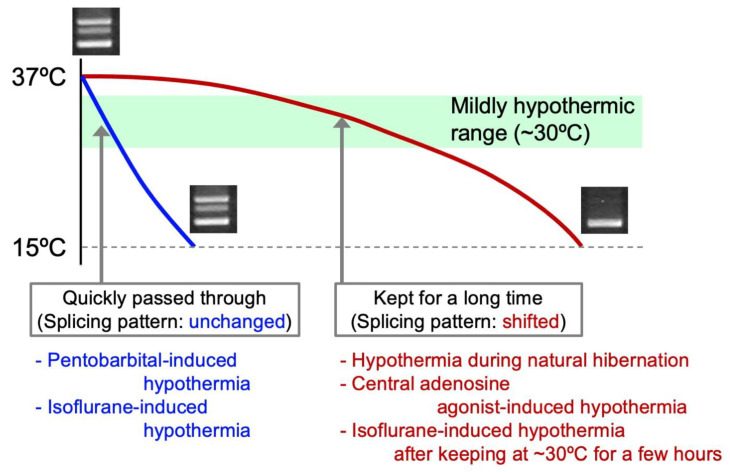
Schema of drop in body temperature contributing to the shift in alternative splicing of *CIRP* in hamsters. The alternative splicing pattern of *CIRP* does not change when body temperature quickly passes through the state of mild hypothermia. On the other hand, maintenance of a state of mild hypothermia for a long time can induce the shift in alternative splicing of *CIRP* as in hibernation.

**Figure 2 ijms-21-07599-f002:**
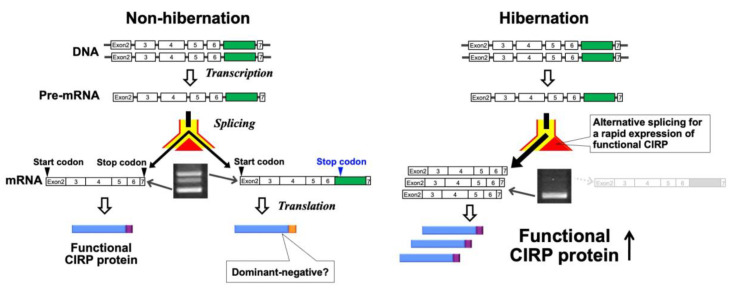
Post-transcriptional regulation of expression of the *CIRP* gene by alternative splicing in non-hibernating euthermic and hibernating hypothermic hamsters. The figure was modified from our published article [14,127].

**Table 2 ijms-21-07599-t002:** Decline rate of body temperature, shift in splicing pattern of CIRP and cardiac state in hamsters and non-hibernators under deep hypothermia.

**Hamsters**				
**Method of Inducing Deep Hypothermia**	**Decline Rate of Body Temperature**	**Shift in Splicing Pattern of CIRP**	**Cardiac Arrhythmias and/or Cardiac Damage**	**References**
Natural hibernation	Slow	Yes	No	[13,128]
Central administration of adenosine A1-receptor agonist combined with cooling	Slow	Yes	No	[128]
Cooling under pentobarbital anesthesia	Rapid	No	Yes	[128]
Cooling under isoflurane anesthesia	Rapid	No	Yes	[13]
Cooling under isoflurane anesthesia after keeping 30 °C for few hours	Slow	Yes (during keeping 30 ºC)	No	[13]
**Non-Hibernators (Mice and Rats)**				
**Method of Inducing Deep Hypothermia**	**Decline Rate of Body Temperature**	**Shift in Splicing Pattern of CIRP**	**Cardiac Arrhythmias and/or Cardiac Damage**	**References**
Central administration of adenosine A1-receptor agonist combined with cooling (Rats)	Slow	Yes	No	[142]
Cooling under isoflurane anesthesia (Mice and Rats)	Rapid (until ~ 20 ºC)	No	Lethal	[15]
Cooling under isoflurane anesthesia after keeping 30 °C for few hours (Mice and Rats)	Slow	Yes	No	[15]

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
