# Peer review of "Temperature-Dependent Alternative Splicing of Precursor mRNAs and Its Biological Significance: A Review Focused on Post-Transcriptional Regulation of a Cold Shock Protein Gene in Hibernating Mammals"

_ijms, 2020, doi:10.3390/ijms21207599_

Round 1
Reviewer 1 Report
This manuscript by Shiina and Shimizu discusses how temperature in affecting alternative RNA splicing and its biological relevance. Although this is a review, emphasis and details are put on the interesting work of the authors on hibernation.
The first part of the manuscript reviews the existing literature and is up-to-date, generally well-written and informative. Although the second part that reviews previous work produced by the authors has been published, some aspects require a more detailed discussion, specifically on the possibility that alternative mRNA stability is involved (see below).
Line 34: Furthermore, alternative splicing can create a functionally…. (required because alternative splicing does not always create inactive variants).
Line 231-232. It is not the molecular mechanisms of genes that are poorly understood, it is the molecular mechanisms regulating their temperature-dependent alternative splicing.
Line 236-237. The impact of RNA secondary structure on splicing concerns not only splice sites but also sequence elements (i.e. exon or intron enhancers and silencers) that affect splice site selection.
Figure 1. A quick and drastic change in temperature (1 h) does not shift the abundance of variants whereas more gradual change (5h) does. Here, I am not convinced that the signal for this change has not occurred in the quick change setting especially if only one hour was given before looking (rather than 1 hour of cold and return to normal temperature for 4 hours). It may take a few hours to allow mRNA turnover to see an accumulation in the production of the short variant (or degradation of the large mRNA variant).
Liner 359. It is a bit odd to start this paragraph with a sentence summarizing the rest of the paragraph. I would keep this sentence for the end of the paragraph.
Line 385. Change the work analyses for considerations
Figure 2. There are three bands in the gel shown on the left. What is the identity of the middle band?
Line 441. The authors claim that they have demonstrated that the shift in CIRP variants is due to alternative splicing. What are the evidence that it is indeed alternative splicing that is affected by temperature and not differential mRNA stability? NMD could be involved because an EJC would be deposited downstream of the premature stop codon in the red exon between exons 6 and 7. Thus, hibernation (low temperature) could promote the NMD-dependent degradation of the long mRNA CIRP variant. A discussion of how to test these hypotheses (splicing or stability) experimentally would also be useful.
Author Response
Responses to the reviewer’s comments (Manuscript ID: ijms-965454)
Responses to Reviewer 1:
Thank you for your useful comments. We modified our manuscript according to your comments. The modified parts and newly added parts are shown in yellow box.
Line 34: Furthermore, alternative splicing can create a functionally…. (required because alternative splicing does not always create inactive variants).
We changed this sentence: “Furthermore, alternative splicing creates a functionally inactive variant,” to “In some cases, alternative splicing creates not only a functionally active variant but also an inactive one,” (page 1, lines 34-35).
Line 231-232. It is not the molecular mechanisms of genes that are poorly understood, it is the molecular mechanisms regulating their temperature-dependent alternative splicing.
We changed this sentence: “their molecular mechanisms are poorly understood” to “molecular mechanisms of temperature-dependent alternative splicing are poorly understood” (page 7, lines 288-289).
Line 236-237. The impact of RNA secondary structure on splicing concerns not only splice sites but also sequence elements (i.e. exon or intron enhancers and silencers) that affect splice site selection.
We added “In addition, the secondary structure of splicing enhancer and silencer motifs also is concerned withsplice site selection” (page 7, lines 293-294).
Figure 1. A quick and drastic change in temperature (1 h) does not shift the abundance of variants whereas more gradual change (5h) does. Here, I am not convinced that the signal for this change has not occurred in the quick change setting especially if only one hour was given before looking (rather than 1 hour of cold and return to normal temperature for 4 hours). It may take a few hours to allow mRNA turnover to see an accumulation in the production of the short variant (or degradation of the large mRNA variant).
Previously, we tried to decrease body temperature of hamsters rapidly and the deep hypothermic condition was maintained for up to 6 hours. However, the splicing pattern was unchanged (reference [15]).
Liner 359. It is a bit odd to start this paragraph with a sentence summarizing the rest of the paragraph. I would keep this sentence for the end of the paragraph.
We moved this sentence to the end of the paragraph (page 10, lines 462-463).
Line 385. Change the work analyses for considerations
We changed it (page 11, line 509).
Figure 2. There are three bands in the gel shown on the left. What is the identity of the middle band?
The middle band would be also one of splicing variants of CIRP. However, sequence analysis is not so far succeeded, probably due to presence of multiple variants in the middle band. Hence, we focused on two major bands.
Line 441. The authors claim that they have demonstrated that the shift in CIRP variants is due to alternative splicing. What are the evidence that it is indeed alternative splicing that is affected by temperature and not differential mRNA stability? NMD could be involved because an EJC would be deposited downstream of the premature stop codon in the red exon between exons 6 and 7. Thus, hibernation (low temperature) could promote the NMD-dependent degradation of the long mRNA CIRP variant. A discussion of how to test these hypotheses (splicing or stability) experimentally would also be useful.
As pointed out, it is possible that NMD-dependent degradation of the long form of CIRP splicing variant might occur during hibernation or/and hypothermia. We added discussion about it (page 13, lines 573-577).

Reviewer 2 Report
This is a very interesting, well-developed, and well-written review that delineates the role of alternative splicing of precursor mRNAs (in particular that of CIRP) during adaptation of plants and animals to changes in environmental and/or body temperature. I did not detect any significant deficiencies that need to be addressed.
Minor points:
- Ln43: should read “reflect” instead of “reflects”.
- Please add “SR” (Ln240) to the list of abbreviations.
- This does not need to be addressed in a revision, but after reading the manuscript I was left wondering whether CIRP (the protein) may bind to its own mRNA (given that CIRP is an RNA-binding protein) to either maintain the dominant-negative effects during “non-hibernation” or regulate the rapid switch to exclusive, functional protein production during onset of hibernation.
Author Response
Responses to the reviewer’s comments (Manuscript ID: ijms-965454)
Responses to Reviewer 2:
Thank you for your useful comments. We modified our manuscript according to your comments. The modified parts and newly added parts are shown in yellow box.
Ln43: should read “reflect” instead of “reflects”.
We changed it (page 1, line 43).
Please add “SR” (Ln240) to the list of abbreviations.
We added “SR” (serine-arginine-rich) to the list of abbreviations (page 6, line 186 and page 13, line 589).
This does not need to be addressed in a revision, but after reading the manuscript I was left wondering whether CIRP (the protein) may bind to its own mRNA (given that CIRP is an RNA-binding protein) to either maintain the dominant negative effects during “non-hibernation” or regulate the rapid switch to exclusive, functional protein production during onset of hibernation..
Thank you for important comment. Your suggestion is very interesting and meaningful. Now, we are doing experiments to clarify functions of CIRP as a mature protein. We would like to examine the suggested hypothesis in future.
